# STNDT: Modeling Neural Population Activity with Spatiotemporal Transformers

**Trung Le**
University of Washington
Seattle, WA
tle45@uw.edu

**Eli Shlizerman**
University of Washington
Seattle, WA
shlizee@uw.edu

## Abstract

Modeling neural population dynamics underlying noisy single-trial spiking activities is essential for relating neural observation and behavior. A recent non-recurrent method - Neural Data Transformers (NDT) - has shown great success in capturing neural dynamics with low inference latency without an explicit dynamical model. However, NDT focuses on modeling the temporal evolution of the population activity while neglecting the rich covariation between individual neurons. In this paper we introduce SpatioTemporal Neural Data Transformer (STNDT), an NDT-based architecture that explicitly models responses of individual neurons in the population across time and space to uncover their underlying firing rates. In addition, we propose a contrastive learning loss that works in accordance with mask modeling objective to further improve the predictive performance. We show that our model achieves state-of-the-art performance on ensemble level in estimating neural activities across four neural datasets, demonstrating its capability to capture autonomous and non-autonomous dynamics spanning different cortical regions while being completely agnostic to the specific behaviors at hand. Furthermore, STNDT spatial attention mechanism reveals consistently important subsets of neurons that play a vital role in driving the response of the entire population, providing interpretability and key insights into how the population of neurons performs computation.[1]

## 1 Introduction

One of the most prominent questions in systems neuroscience is how neurons perform computations that give rise to behaviors. Recent evidence suggests that computation in the brain could be governed at the population level [1, 2]. Population of neurons are proposed to obey an internal dynamical rule that drives their activities over time [3, 4]. Inferring these dynamics on a single trial basis is crucial for understanding the relationship between neural population responses and behavior, subsequently enabling the development of robust decoding schemes with wide applicability in brain-computer interfaces (BCI) [5–7]. However, modeling population dynamics on single trials is challenging due to the stochasticity of individual neurons making their spiking activity vary from trial to trial even when they are subject to identical stimuli or recorded under repeated behavior conditions.

A direct approach to reduce the trial-to-trial variability of neural responses could be to average responses over repeated trials of the same behavior [8, 9], to convolve the neural response with a Gaussian kernel [10], or in general, to define a variety of neural activity measures [11]. However, more success was found in approaches that explicitly model neural responses as a dynamical system, including methods treating the population dynamics as being linear [12, 13], switched linear [14], non-linear [15, 16], or reduced projected nonlinear models [11]. Recent approaches leveraging

---

[1]Code is available at https://github.com/shlizee/STNDT

36th Conference on Neural Information Processing Systems (NeurIPS 2022).

recurrent neural networks (RNN) have shown promising progress in modeling distinct components of a dynamical system - neural latent states, initial conditions and external inputs - on a moment-to-moment basis [15, 17, 18]. These sequential methods rely on continuous processing of neural inputs at successive timesteps, causing latency that hampers applicability in real-time decoding of neural signals. Consequently to RNN-based approaches, Neural Data Transformer (NDT) [16] was proposed as a non-recurrent approach to improve inference speed by leveraging the transformers architecture which learns and predicts momentary inputs in parallel [19]. While successful, NDT has only focused on modeling the relationship of neural population activity between timesteps while ignoring the rich covariation among individual neurons. Neurons in a population have been shown to have heterogeneous tuning profiles where each neuron has a different level of preference to a particular muscle movement direction [20, 21]. Neuron pairs also exhibit certain degree of correlation in terms of trial-to-trial variability (noise correlation) that affects the ability to decode the behaviors they represent [2, 22]. These spatial correlations characterize the amount of information that can be encoded in the neural population [22], necessitating the need to model the neural population activity across both time and space dimensions.

In this work, we propose to incorporate the information distributed along the spatial dimension to improve the learning of neural population dynamics, and introduce *SpatioTemporal* Neural Data Transformer, an architecture based on Neural Data Transformer which explicitly learns both the spatial covariation between individual neurons and the temporal progression of the entire neural population. We summarize our main contributions as follows:

- We introduce STNDT which allows the transformer to learn both the spatial coordination between neurons and the temporal progression of the population activity by letting neurons attend to each other while also attending over temporal instances.

- We propose a contrastive training scheme, complementary to the mask modeling objective, to ensure the robustness of model prediction against induced noise augmentations.

- We validate our model's performance on four neural datasets in the publicly available Neural Latents Benchmark suite [23] and show that ensemble variants of our model outperforms other state-of-the-art methods, demonstrating its capability to model autonomous and non-autonomous neural dynamics in various brain regions while being agnostic to external behavior task structures.

- We show that the spatial attention, a feature unique to STNDT, identifies consistently important subsets of neurons that play an essential role in driving the response of the entire population. This exclusive attribute of STNDT provides interpretability and key insights into how the neural population distributes the computation workload among the neurons.

## 2   Related Work

**Modeling spatial covariation in neural population:** Neurons act as an orchestrated system which collectively encodes behaviors in a distributed and redundant manner. Many previous works have studied and incorporated neural variability across neurons to closely match firing statistics observed in multi-channel neural recordings [24–30]. [25] simulated population responses within a Dichotomized Gaussian framework and solved for signal and noise correlations numerically. [26, 27] developed Generative Adversarial Networks that were able to capture pairwise correlations among the neurons and generate realistic firing patterns. [28–30] modeled the population responses as being generated from a latent variable with learnable covariance matrix reflecting covariability among the neurons.

While these methods resemble our work in the overarching motivation of capturing interactions among neurons, they rely on the knowledge of the respective stimuli/conditions that the trials belong to when modeling the interaction. On the other hand, STNDT is trained in an unsupervised manner and learns the rich covariation among neurons encompassing all recorded behaviors without access to any external observation apart from the population spiking activity. In addition, while the goal of aforementioned methods is to generate realistic firing activities associated with induced stimuli, oftentimes with some assumptions regarding their statistics (e.g. noise correlation is shared across time bins and trials), STNDT aims to uncover the denoised firing patterns behind the noisy single-trial spiking activity and does not depend on any prior assumptions regarding their firing statistics.

**Transformers for modeling spatiotemporal data:** Transformers were initially developed to model the relationship between words in a sentence, which can be thought of as a temporal progression of a

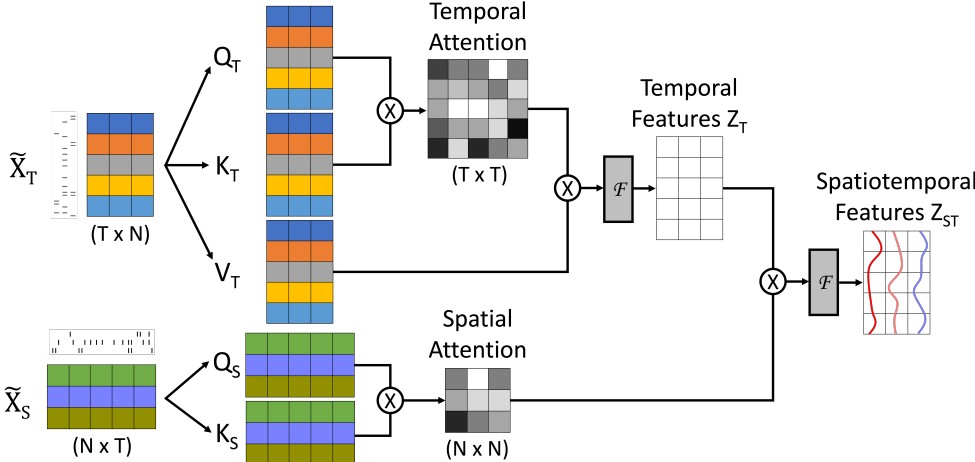

Figure 1: Spatiotemporal Neural Data Transformer (STNDT) architecture. Separate multihead self-attention modules are trained to learn spatial covariation and temporal progression of neural activities. Temporal attention feature matrix is treated as the matrix V upon which spatial attention is multiplied to give the final spatiotemporal features. Colors represent entities over which self attention is performed. The complete STNDT consists of multiple layers of such spatiotemporal attention modules.

sequence of tokens. Recent works have leveraged the self-attention mechanism in transformers to model spatiotemporal data types where there exist an additional interacting dimensions possessing distinct dynamics, such as trajectories of traffic agents [31–33], dynamic scene graph of video [34], or 3D human motion [35]. However, in these works the spatial interaction at each timestep and the temporal dynamics for each entity are captured independently, treating the other dimension as the batch dimension at each attention block. In contrast, STNDT interleaves spatial and temporal attention in a unified framework, using spatial attention to re-weight temporal features and enabling direct study of each individual neuron's role in driving the population dynamics.

**Interpretability of self-attention mechanism:** Several approaches have been proposed to probe the inner workings of black-box deep learning models [36–38]. Unlike our work, these approaches attempted to attribute importance of visual inputs to the model prediction in a supervised setting and did not take into account interaction between input features. For attention-based models, the weights of attention matrix have been used as a tool to provide certain level of interpretability [39–42]. The interpretability is built upon the fact that attention weights signify how much influence other inputs have on a particular input in deciding its final outcome in a self-supervision manner. This influence might align with some human interpretable meaning, such as linguistic patterns [43]. In our work, we further leverage attention weights to gain insights into the interaction of neurons from multi-channel neural recordings.

## 3 Methods

**Problem formulation:** Single-trial spiking activity of a neural population can be represented as a spatiotemporal matrix $X \in \mathbb{N}^{T \times N}$, where each column $X_i \in \mathbb{N}^T$ is the time series of one neuron, $T$ is the number of time bins for each trial, and $N$ is the number of neurons in the population. Each element $X_{tn}$ in the matrix is the number of action potentials (spikes) that neuron $n$ fires within the time bin $t$. Spike counts are assumed to be samples of an inhomogeneous Poisson process $P(\lambda(t, n))$ where $\lambda(t, n)$ is the underlying true firing rate of neuron $n$ at time $t$. The matrix $Y \in \mathbb{R}^{T \times N}$ containing $\lambda(t, n)$ fully represents the dynamics of the neural population and explains the observable spiking data of the respective trial. We propose to learn the mapping $\phi(X; W) : X \to Y$ by the Spatiotemporal Transformer with the set of weights $W$.

**Spatiotemporal Neural Data Transformer:** At the core of the transformer architecture is the multihead attention mechanism, where feature vectors learn to calibrate the influence of other feature

vectors in their transformation. Spike trains are embedded into feature matrices $\tilde{X}$ with added sinusoidal positional encoding to preserve order information as initially proposed in [19]. We employed separate embeddings to encode positions in each temporal and spatial dimension individually, resulting in two distinct feature embeddings $\tilde{X}_T = Emb(X) + P_T$ and $\tilde{X}_S = Emb(X^\top) + P_S$.

A set of three matrices $W_T^Q$, $W_T^K$, $W_T^V \in \mathbb{R}^{N \times N}$ are learned to transform $T$ $N$-dimensional embedding $\tilde{X}_T = \{\tilde{x}_1, \tilde{x}_2, ..., \tilde{x}_T\}$ to queries $Q_T = \tilde{X}_T W_T^Q$, keys $K_T = \tilde{X}_T W_T^K$ and values $V_T = \tilde{X}_T W_T^V$, upon which latent variable $Z_T$ is computed as:

$$Z_T = \text{Attention}(Q_T, K_T, V_T) = \mathcal{F}\left(\text{softmax}\left(\frac{Q_T K_T^\top}{\sqrt{N}}\right) V_T\right) \qquad (1)$$

The outer product of $Q_T K_T^\top$ represents the attention each $x_i$ pays to all other $x_j$ and determines how much influence their values $v_j$ have on its latent output $z_i$. $\mathcal{F}$ is the sequence of concatenating multiple heads and feeding through a feedforward network with ReLU activation [19]. We used 2 heads for all reported models.

Implementations of transformers in popular applications such as in natural language processing literature consider each feature vector $x_i$ as an $N$-dimensional token in a sequence, equivalent to a word in a sentence. Elements in the $N$-dimensional vector therefore serve as a convenient numerical representation and do not have inherent relationships among them. The attention mechanism thus only models the relationship between tokens in a sequence. In our application, each feature vector $x_i$ is a collection of firing activities of $N$ physical neurons among which there exists an interrelation as neuronal population acts as a coordinated structure with complex interdependencies rather than standalone individuals. We therefore propose to model both the temporal relationship - the evolution of neural activities - and the spatial relationship - covariability of neurons - by learning two separate multihead attention blocks (Figure 1). The temporal latent state $Z_\mathcal{T}$ is computed with temporal attention block as in Equation 1. In parallel, spatial attention block operates on the spatial embedding $\tilde{X}_S$ and learns an attention weights matrix signifying the relationship between neurons:

$$A_\mathcal{S} = \text{softmax}\left(\frac{Q_\mathcal{S} K_\mathcal{S}^\top}{\sqrt{T}}\right) \qquad (2)$$

where $Q_\mathcal{S} = \tilde{X}_S W_\mathcal{S}^Q$ and $K_\mathcal{S} = \tilde{X}_S W_\mathcal{S}^K$.

This $A_\mathcal{S}$ matrix is then multiplied with the transpose of temporal latent state $Z_\mathcal{T}$ to incorporate the influence of spatial attention on the final spatiotemporal latent state $Z_\mathcal{ST}$:

$$Z_\mathcal{ST} = \mathcal{F}(A_\mathcal{S} Z_\mathcal{T}^\top) \qquad (3)$$

For stable training, as in [19] we used layer normalization before $\tilde{X}_T$, $\tilde{X}_S$, $A_S Z_T^\top$ and feedforward layers. Residual connections are also employed around temporal attention, feedforward layers and $A_S Z_T^\top$.

**Mask modeling and contrastive losses:** Similar to [16], we train the spatiotemporal transformer in an unsupervised way with BERT's mask modeling objective [44]. During training, a random subset of spike bins along both spatial and temporal axes of input $X$ are masked (zero-ed out or altered) and the transformer is asked to reconstruct the log firing rate at the masked bins such that the Poisson negative log likelihood is minimized:

$$\mathcal{L}_{mask} = \sum_{i=1}^{N} \sum_{j=1}^{T} \exp(\tilde{z}_{ij}) - \tilde{x}_{ij}\tilde{z}_{ij} \qquad (4)$$

where $\tilde{z}_{ij}$ and $\tilde{x}_{ij}$ are the log output firing rate and input spike of neuron $i$ at timestep $j$ if location $ij$ is masked.

Neural dynamics are shown to be embedded in a low-dimensional space, i.e. model prediction should be fairly consistent when a smaller subset of neurons are used compared to when the entire population is taken into account. Furthermore, in stereotyped behaviors often found in neuroscience experiments, trials with the same condition should yield similar output firing rate profiles. Therefore, to enhance robustness of model prediction to neural firing variability we further constrain model firing rate outputs by a contrastive loss, such that different augmentations of the same trial input remain closer

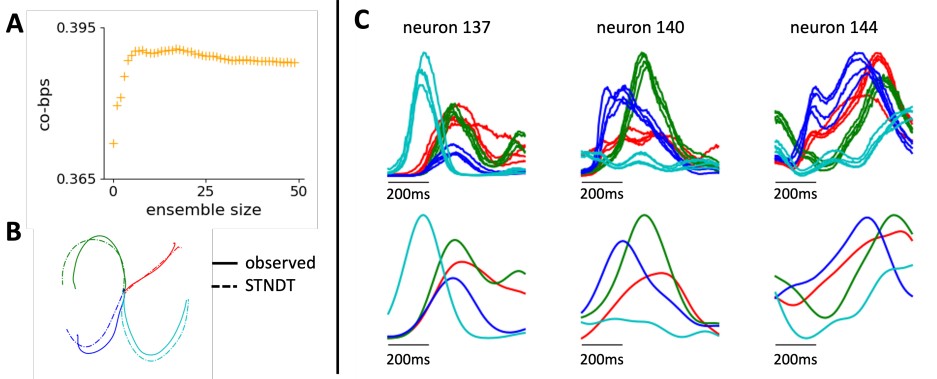

Figure 2: **A**: co-bps metrics improves when multiple models are ensembled together. **B**: STNDT facilitates accurate inference of behavior from spiking data. Decoded hand trajectories from 4 trials (dashed line) closely match the ground truth trajectories (solid line). **C**: STNDT uncovers the stereotyped feature of neural activity in structured behaviors. Firing rate prediction and PSTHs of three example neurons are shown. Trials belonging to the same condition are plotted with the same color (4 trials per condition shown). All results are shown for MC_Maze dataset.

to each other and stay distant to other trial inputs. We adopt the NT-XEnt contrastive loss introduced in [45]:

$$\mathcal{L}_{contrastive} = \sum_{ij} l_{ij} = \sum_{ij} -\log \frac{\exp(\text{sim}(z_i, z_j)/\tau)}{\sum_{k=1}^{2N} \mathbf{1}_{k \neq i} \exp(\text{sim}(z_i, z_k)/\tau)} \tag{5}$$

where $\text{sim}(u, v) = u \top v / (\|u\| \|v\|)$ is the cosine similarity between two predictions $u$ and $v$ on two different augmentations of input $x$ and $\tau$ is the temperature parameter.

Transformations such as dropping out neurons and jittering samples in time have been used to create different views of neural data [46]. In our work, we define the augmentation transformation as random dropout and alteration of spike counts at random elements in the original input matrix $X$, similar to how masking is done, i.e. zero out or change spike counts to random integers at random neurons and timesteps. See Appendix for details on probabilities used to create these augmentations.

**Bayesian hyperparameter tuning:** We follow [47] to use Bayesian optimization for hyperparameters tuning. We observe that the primary metrics co-smoothing bits/spike (co-bps) are not well correlated with the mask loss (see Figure 1 in the Appendix , while co-bps, vel $R^2$, psth $R^2$ and fp-bps are more pairwise correlated. Therefore, we run Bayesian optimization to optimize co-bps for $M$ models then select the best $N$ models as ranked by validation co-bps, and ensemble them by taking the mean of the predicted rates of these $N$ models.

## 4 Experiments and results

**Datasets and evaluation metrics:** We evaluate our model performance on four neural datasets in the publicly available Neural Latents Benchmark [23]: MC_Maze, MC_RTT, Area2_Bump, and DMFC_RSG. The 4 datasets cover autonomous and non-autonomous neural population dynamics recorded on rhesus macaques in a variety of behavioral tasks (delayed reaching, self-paced reaching, reaching with perturbation, time interval reproduction) spanning multiple brain regions (primary motor cortex, dorsal premotor cortex, somatosensory cortex, dorso-medial frontal cortex). The diverse scenarios and systems offer comprehensive evaluation of a latent variable model and serve as a standardized benchmark for comparison between different modeling approaches. We use different metrics to measure performance of our model depending on the particular behavior task of each dataset, following the standard evaluation pipeline in [23]. We evaluate and report our model performance on the hidden test split held by NLB to have a fair comparison with other state-of-the-art (SOTA) methods. See [23] for further details of evaluation strategy and how the metrics are calculated.

Table 1: Performance of STNDT as compared to SOTA methods on MC_Maze and MC_RTT datasets

| Methods | MC_Maze | | | | MC_RTT | | |
|---|---|---|---|---|---|---|---|
| | co-bps↑ | vel $R^2$↑ | psth $R^2$↑ | fp-bps↑ | co-bps↑ | vel $R^2$↑ | fp-bps↑ |
| GPFA | 0.1872 | 0.6399 | 0.5150 | – | 0.1548 | 0.5339 | – |
| Smoothing | 0.2109 | 0.6238 | 0.1853 | – | 0.1468 | 0.4142 | – |
| SLDS | 0.2249 | 0.7947 | 0.5330 | 1.1579 | 0.1649 | 0.5206 | 0.0620 |
| MINT | 0.3304 | **0.9121** | **0.7496** | 0.2076 | 0.1676 | 0.5953 | 0.1012 |
| AutoLFADS | 0.3364 | 0.9097 | 0.6360 | 0.2349 | 0.1868 | 0.6167 | 0.1213 |
| iLQR-VAE | 0.3559 | 0.8840 | 0.6062 | 0.1480 | – | – | – |
| AESMTE1 (single) | 0.3599 | 0.9105 | 0.6641 | 0.2470 | 0.1927 | **0.6627** | 0.1229 |
| AESMTE3 (ensemble) | 0.3676 | 0.9114 | 0.6683 | 0.2589 | 0.2053 | 0.6334 | **0.1344** |
| STNDT single (ours) | 0.3691 | 0.8985 | 0.6567 | 0.2505 | 0.1938 | 0.6143 | 0.0988 |
| STNDT ensemble (ours) | **0.3862** | 0.9095 | 0.6693 | **0.2686** | **0.2095** | 0.6270 | 0.1244 |

Table 2: Performance of STNDT as compared to SOTA methods on Area2_Bump and DMFC_RSG datasets

| Methods | Area2_Bump | | | | DMFC_RSG | | | |
|---|---|---|---|---|---|---|---|---|
| | co-bps↑ | vel $R^2$↑ | psth $R^2$↑ | fp-bps↑ | co-bps↑ | tp-corr↓ | psth $R^2$↑ | fp-bps↑ |
| GPFA | 0.1680 | 0.5975 | 0.5289 | – | 0.1176 | −0.3763 | 0.2142 | – |
| Smoothing | 0.1544 | 0.5736 | 0.2084 | – | 0.1202 | −0.5139 | 0.2993 | – |
| SLDS | 0.1960 | 0.7385 | 0.5740 | 0.0242 | 0.1243 | −0.5412 | 0.3372 | −0.0418 |
| MINT | 0.2735 | 0.8877 | **0.9135** | 0.1483 | 0.1821 | −0.6929 | **0.7013** | 0.1650 |
| AutoLFADS | 0.2569 | 0.8492 | 0.6318 | 0.1505 | 0.1829 | **−0.8248** | 0.6359 | 0.1844 |
| iLQR-VAE | – | – | – | – | – | – | – | – |
| AESMTE1 (single) | 0.2801 | 0.8675 | 0.6367 | 0.1523 | 0.1733 | −0.6189 | 0.5267 | 0.1511 |
| AESMTE3 (ensemble) | 0.2860 | **0.8999** | 0.7109 | **0.1603** | 0.1886 | −0.7601 | 0.6064 | 0.1828 |
| STNDT single (ours) | 0.2818 | 0.8766 | 0.6454 | 0.1357 | 0.1859 | −0.5205 | 0.6051 | 0.1601 |
| STNDT ensemble (ours) | **0.2898** | 0.8913 | 0.7368 | 0.1476 | **0.1940** | −0.4857 | 0.6452 | **0.1910** |

- **Co-smoothing (co-bps)**: the primary metric, measuring the ability of the model to predict activity of held-out neurons it has not seen during training. Co-bps is tied to the goodness of mask loss evaluated for held-out neurons.

- **Behavior decoding (vel $R^2$ or tp-corr)**: measures how useful the model firing rates prediction can be used to decode behavior (the velocity of primate's hand in the cases of MC_Maze and Areas_Bump datasets, or the correlation between neural speed and time between Set cue and Go response in DMFC_RSG dataset).

- **Match to peri-stimulus time histogram (psth $R^2$)**: indicates how well predicted firing rates match the peri-stimuls time histogram in repeated, stereotyped task structures.

- **Forward prediction (fp-bps)**: measures model's ability to predict unseen future activity of the neural population. It is computed in the similar manner as co-bps but on the held-out time points of all neurons.

**Baselines:** We compare STNDT against the following baselines, all of which have been evaluated using the same held-out test split.

- **Smoothing** [23]: A simple method where a Gaussian kernel is convolved with held-in spikes to produce smoothed held-in firing rates. Then a Poisson Generalized Linear Model (Poisson GLM) is fitted from the held-in smoothed rates to held-out rates.

- **GPFA** [10]: extracts population latent states as a smooth and low dimensional evolution by combining smoothing and dimension reduction in a common probabilistic framework.

- **SLDS** [14]: models neural dynamics as a switching linear dynamical system, which breaks down nonlinear data into sequences of simpler dynamical modes.

- **AutoLFADS** [17]: models population activity as a non-linear dynamical system with bi-directional recurrent neural networks at the core and a scalable framework of hyperparameter tuning.

- **MINT** [48]: an interpretable decode algorithm that exploits the sparsity and stereotypy of neural activity to interpolate neural states using a library of canonical neural trajectories.

- **iLQR-VAE** [49]: improves upon LFADS with iterative linear quadratic regulator algorithm, an optimization-based recognition model to replace RNN as the inference network.

- **NDT** [16]: leverages transformer architecture with some adaption to neural data to model temporal progression of neural activity across time. AESMTE1 is the best single model and AESMTE3 is the best emsemble of multiple models found as a result of Bayesian hyperparameter tuning [47].

### 4.1 Spatiotemporal transformer achieves state-of-the-art performance in modeling autonomous dynamics

We first tested STNDT on recordings of dorsal premotor (PMd) and motor cortex (M1) of a monkey performing a delayed reaching task (MC_Maze dataset) to evaluate the ability of STNDT to uncover single-trial population dynamics in a highly structured behavior. The dataset has been studied extensively in previous work [15–17]. It consists of 2869 trials of monkey performing a center-out reaching task in a maze with obstructing barriers, composing 108 different conditions for straight and curved reaching trajectories. The monkey is trained to hold the cursor at the center while the target is presented and only move the cursor to reach the target after a 'Go' cue. The neural dynamics during the preparation and execution periods is well modeled as an autonomous dynamical system [15].

We observed that by explicitly modeling spatial interaction, STNDT outperformed other state-of-the-art methods and improved NDT's ability to model autonomous single-trial dynamics as measured by the negative log likelihood of unobserved neural activity. The single STNDT model improved both Poisson log likelihood of heldout neurons (co-bps) and heldout timesteps (fp-bps). The performance is further increased by aggregating multiple STNDT models as shown in Table 1 and Figure 2A.

Since MC_Maze features repeated trials, the prediction of any latent variable models should uncover stereotypical patterns of neuronal responses for trials belonging to the same condition. Therefore, we computed PSTH which is the average of neural population response across trials of the same condition, and measure $R^2$ matching of model prediction to this PSTH. We observed that with the help of spatial modeling and contrastive loss, STNDT boosts NDT ability to recover this stereotyped firing pattern 1. We show in Figure 2C several responses of example neurons. STNDT firing rates prediction of trials under the same condition exhibit a consistent, stable PSTH as desired. These predicted rates also decode behaviors accurately when mapped to hand velocity via a linear regression model (Table 1, Figure 2B).

### 4.2 Spatiotemporal transformer improves inference of non-autonomous neural dynamics underlying naturalistic behaviors

There is much interest in systems neuroscience to study neural dynamics in unconstrained, naturalistic behaviors as it is crucial for developing ubiquitous BCI decoders. We evaluated STNDT's applicability to this setting via recordings in primary motor cortex during self-paced reaching task (MC_RTT dataset) [23, 50]. Unlike MC_Maze dataset, the monkey in this task continuously acquires targets which appear randomly in an 8x8 grid without preparatory periods, resulted in a wide variety of hand trajectories and trial lengths. We observe that STNDT achieves SOTA performance on the primary metric co-bps and performs on par with NDT on remaining metrics, while maintaining a more robust performance against random initializations of model weights (Table 1 and Appendix).

### 4.3 Spatiotemporal transformer better captures input-driven dynamics underlying sensory processes

We next tested STNDT in a setting where unexpected input perturbations affect the neural dynamics in somatosensory cortex to probe whether STNDT can leverage spatial interaction to improve modeling of non-autonomous dynamics in this brain region. Area2_Bump dataset consists of recordings from the Area 2, which was shown in previous works to be driven by mechanical perturbation to the arm and contains information about whole-arm kinematics [23, 51]. The task comprises of active and passive trials with a center hold period at the start. During active trials, the monkey performs a classic

center-out reaching task. In passive trials, a force is applied on the monkey's hand in a random direction via a manipulandum, after which the monkey has to return to the center target and proceed with the task as in active trials. Despite the relatively small scale of the dataset, STNDT brings about further improvements to NDT performance in terms of co-bps and psth-$R^2$, on both single and ensemble levels.

### 4.4 Spatiotemporal transformer enhances prediction of neural population activity during cognitive task

Dorsomedial frontal cortex (DMFC) is believed to serve as an intermediate layer between low-level sensory and motor areas, and possess distinct confluence of internal dynamics and inputs [52, 53]. We are therefore interested to see if characterizing spatial relationship alongside temporal relationship and incorporating contrastive loss could help STNDT better model the dynamics in this brain region. We tested STNDT on the DMFC_RSG dataset [23, 53] consisting of recordings from a rhesus macaque performing a time-interval reproduction task. The monkey is presented two 'Ready' and 'Set' stimuli separated by a specific time interval $t_s$ while fixating eye and hold the joystick at the center position. It then has to execute a 'Go' response by either an eye saccade or joystick movement such that the time interval $t_p$ between its reponse and the 'Set' cue is sufficiently close to $t_s$. STNDT successfully captures the dynamics in this cognitive task, outperforming NDT by a large margin across co-bps, psth-$R^2$ and fp-bps on both single and ensemble level (Table 2).

### 4.5 Spatial attention mechanism identifies important subsets of neurons driving the population dynamics

In Figure 3, we visualize spatial attention weights obtained from STNDT on the MC_Maze dataset in the first and last attention layers. Attention map for remaining datasets are provided in Appendix. Interestingly, spatial attention shows that in early layers, only a small subsets of neurons in the population are consistently attended to by all neurons. The spatial attention tends to disperse as the model goes to deeper layers. Strikingly, the subset of heavily-attended neurons stays relatively identical across different trials, hinting that these neurons might play a crucial role in driving the population response to the behavior task. We further tested this hypothesis by incrementally dropping the neurons heavily attended to (i.e. zeroing out their spiking activity input to the model) in a descending order of their attention weights identified in the first layer. We observed that dropping these important neurons identified by STNDT caused a significant decline in the model performance (Figure 4). The performance decline was significantly more than the case where the same number of random neurons are dropped. To rule out the possible case that dropping neurons only has adverse effect on the spatial attention module but that effect propagates to the subsequent modules and indirectly impacts the performance of the overall STNDT pipeline, we repeated the experiment on the vanilla NDT model which, unlike STNDT, lacks a spatial attention structure. Interestingly, we observed the same performance deterioration when we dropped the spiking activity of STNDT-identified important neurons and asked a pretrained vanilla NDT to make inference on the resulting inputs. This finding suggests that the impact of the important neurons that only STNDT can identify might potentially generalize to other latent variable models that without input from these neurons, some latent variable models might not function optimally. We provide additional results from similar analyses on GPFA and Smoothing models in the Appendix.

We further examine whether important neurons were selected by the spatial attention mechanism based on some criteria more sophisticated than simple firing statistics, as more active neurons tend to have higher signal-to-noise ratio and might encode more useful information with regard to behaviors. We find that the important neurons are not the ones with the highest spike counts or the least variability in spiking activity. In fact, attention weights of a neuron do not correlate or only correlate weakly to its firing activity statistics, as we show in Table 3 the Pearson's correlation of a neuron's attention weight with the mean and variance of its spiking activity. All correlation values have $p$-value < 1e-4. These results indicate that STNDT's spatial attention has picked up on meaningful population features that are more significant than firing statistics of the neurons.

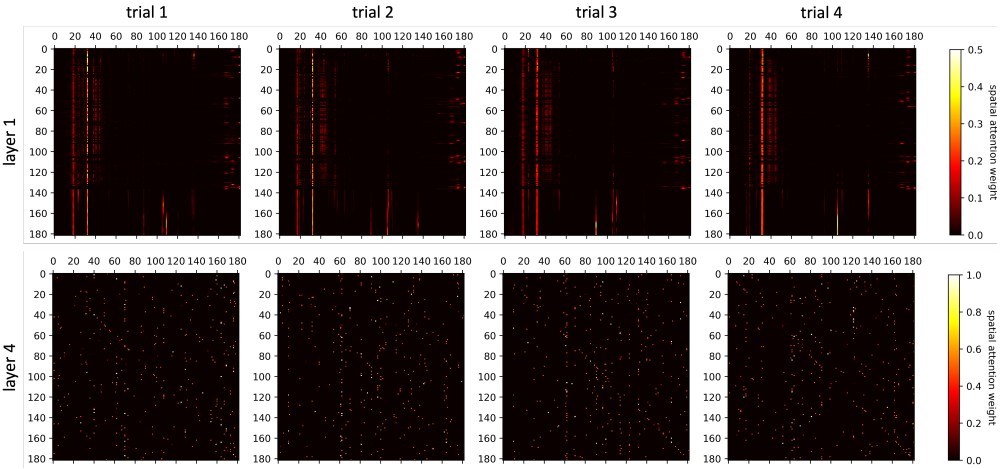

Figure 3: Visualization of STNDT's spatial attention weights in the first and last layers of four example trials. Attention weights in layer 1 reveal a consistent subset of neurons that are heavily attended to by all neurons in the population. The attention becomes more dispersed in deeper layers. Results are shown for 182 neurons in MC_Maze dataset.

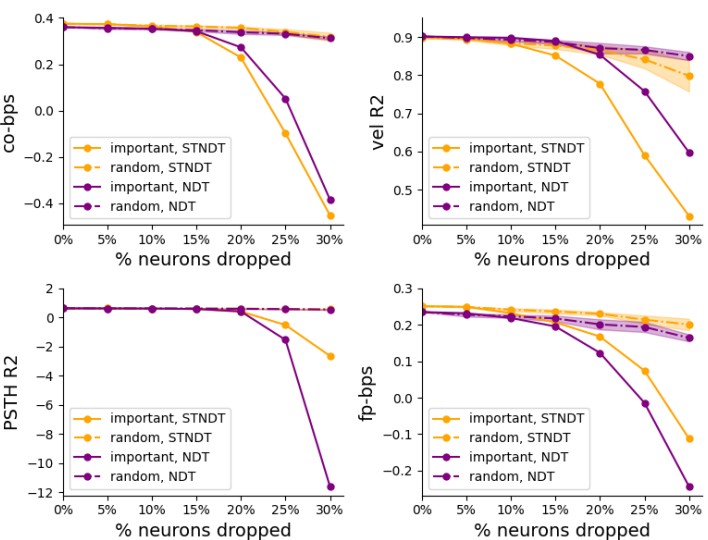

Figure 4: Spatial attention module, unique to STNDT, identifies important neurons that are the main driving force of population response to behavioral task. Performance of STNDT as measured by four evaluation metrics are plotted as neurons are incrementally dropped from input neural population. Performance significantly deteriorates when important neurons identified by STNDT are dropped, while only decreases slightly when random neurons are dropped. The effect of important neurons indentified by STNDT generalizes to vanilla NDT, which lacks a spatial attention structure. Shaded region represents 2 standard error of the mean. Results are shown for MC_Maze dataset.

Table 3: Pearson's correlation between spatial attention weight of a neuron versus mean and variance of its spiking activity.

|  | MC_Maze | MC_RTT | Area2_Bump | DMFC_RSG |
|---|---|---|---|---|
| $\rho$(spike mean, attn weight) | 0.0164 | 0.2217 | 0.0327 | 0.0852 |
| $\rho$(spike var, attn weight) | 0.0124 | 0.2189 | 0.0353 | 0.0937 |

Table 4: Ablation Study: Performance of STNDT on MC_Maze and MC_RTT datasets with and without contrastive loss (CL) on single and ensemble levels.

| Methods | MC_Maze | | | | MC_RTT | | |
|---|---|---|---|---|---|---|---|
| | co-bps↑ | vel $R^2$↑ | psth $R^2$↑ | fp-bps↑ | co-bps↑ | vel $R^2$↑ | fp-bps↑ |
| AESMTE1 (single) | 0.3599 | 0.9105 | 0.6641 | 0.2470 | 0.1927 | 0.6627 | 0.1229 |
| AESMTE3 (ensemble) | 0.3676 | 0.9114 | 0.6683 | 0.2589 | 0.2053 | 0.6334 | 0.1344 |
| STNDT single w/o CL | 0.3668 | 0.8979 | 0.6549 | 0.2471 | 0.1865 | 0.5988 | 0.0964 |
| STNDT single w/ CL | 0.3691 | 0.8985 | 0.6567 | 0.2505 | 0.1938 | 0.6143 | 0.0988 |
| STNDT ensemble w/o CL | 0.3843 | 0.9090 | 0.6686 | 0.2675 | 0.2065 | 0.6352 | 0.1260 |
| STNDT ensemble w/ CL | 0.3862 | 0.9095 | 0.6693 | 0.2686 | 0.2095 | 0.6270 | 0.1244 |

Table 5: Ablation Study: Performance of STNDT on Area2_Bump and DMFC_RSG datasets with and without contrastive loss (CL) on single and ensemble levels.

| Methods | Area2_Bump | | | | DMFC_RSG | | | |
|---|---|---|---|---|---|---|---|---|
| | co-bps↑ | vel $R^2$↑ | psth $R^2$↑ | fp-bps↑ | co-bps↑ | tp-corr↓ | psth $R^2$↑ | fp-bps↑ |
| AESMTE1 (single) | 0.2801 | 0.8675 | 0.6367 | 0.1523 | 0.1733 | −0.6189 | 0.5267 | 0.1511 |
| AESMTE3 (ensemble) | 0.2860 | 0.8999 | 0.7109 | 0.1603 | 0.1886 | −0.7601 | 0.6064 | 0.1828 |
| STNDT single w/o CL | 0.2765 | 0.8773 | 0.7169 | 0.1498 | 0.1824 | −0.5059 | 0.6134 | 0.1473 |
| STNDT single w/ CL | 0.2818 | 0.8766 | 0.6454 | 0.1357 | 0.1859 | −0.5205 | 0.6051 | 0.1601 |
| STNDT ensemble w/o CL | 0.2904 | 0.8937 | 0.7303 | 0.1491 | 0.1931 | −0.5186 | 0.6429 | 0.1888 |
| STNDT ensemble w/ CL | 0.2898 | 0.8913 | 0.7368 | 0.1476 | 0.1940 | −0.4857 | 0.6452 | 0.1910 |

## 4.6 Ablation Study: Contrastive loss encourages consistency of model prediction and improves performance

We conduct an ablation study to assess the effectiveness of contrastive loss on the overall performance of STNDT. Tables 4 and 5 report how the model scores on different metrics across all four datasets on the single and ensemble levels. In general, we observe that having contrastive loss further improves the performance of STNDT on predicting neural activity of heldout neurons (co-bps) and heldout timesteps (fp-bps). The contribution of contrastive loss is most eminent on MC_Maze dataset.

## 5 Discussion

In this paper we presented STNDT, a novel architecture based upon NDT [16] that explicitly learns the covariation among individual neurons in the population alongside the momentary evolution of the population spiking activity in order to infer the underlying firing rates behind highly variable single-trial spike trains. By incorporating self-attention along both spatial and temporal dimensions as well as a contrastive loss, STNDT enhances NDT's ability to model dynamics spanning a variety of tasks and brain regions, most notably by the accurate prediction of activity for unseen neurons (co-bps). Although STNDT does not consistently outperform NDT on other secondary metrics, we show in the Appendix that STNDT is more robust to random initializations and performs better than NDT on average across random seeds. Moreover, the improvement STNDT contributes on co-bps is the direct reflection of the spatial attention's success. Since the spatial attention module aims to learn the relationship between all (observed and unobserved) neurons at training time, it will leverage this information to infer activities of unobserved neurons based on those of observed neurons at testing time, which is exactly what co-bps measures. Finally, the novel spatial attention mechanism unique to STNDT brings about valuable interpretability as it discovers influential subsets of neurons whose activities contain salient information about the response of the entire neural population without which some latent variable models might not function optimally.

**Acknowledgment:** This work was supported in part by National Science Foundation grant OAC-2117997 and Washington Research Fund to ES. Authors also acknowledge the partial support by the Departments of Electrical Computer Engineering (TL and ES), Applied Mathematics (ES), the Center of Computational Neuroscience (ES), and the eScience Center (ES) at the University of Washington.

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
