# OpenReview forum: "STNDT: Modeling Neural Population Activity with Spatiotemporal Transformers"
_NeurIPS.cc/2022/Conference — NeurIPS 2022 Accept_

### Official Review · Reviewer_gfnD · 2022-07-06

**Rating:** 4
**Confidence:** 4
**Soundness:** 3 good
**Presentation:** 3 good
**Contribution:** 2 fair

**Summary:**

This paper addresses the task of modelling noisy single-trial spiking activities and extends the Neural Data Transformer (NDT) model by including an attention mechanism between neurons that treats each neuron as a token. Additionally, the authors add the SimCLR contrastive learning objective to the model trying to improve generalization. The paper investigates the new model’s performance on spiking neuron activity of four publicly available motor control datasets.

**Questions:**

 1. Tables 1 + 2: I do not see an overall improvement by your additional attention model over AESMTE1/3. The metrics are above the baseline as frequently as they are below i, suggesting that we're looking at random fluctuations. Am I missing something?

 1. Tables 3 + 4: Again, the ablation study seems to suggest to me that the contrastive objective does not actually improve the model. Am I missing something?

 1. Does the model use positional encodings? It seems like that would be a useful thing to encode properties of the tokens such as neuron identity. Otherwise the tokens are permutation invariant, which means the attention mechanism has to infer the token identity from the current spatial (or temporal) pattern of spikes, which probably limits the model quite substantially (and unnecessarily). Please explain whether you used positional encodings and, if so, how or, if not, why.

 1. L 102ff.: Please describe in more detail how exactly you perform the masking. Do you drop individual entries X_ij or entire rows and/or columns?

 1. L 119: Please expand on how exactly you alter the spike counts for the contrastive loss and provide the dropout probability.

 1. Fig 4: It is not clear to me what the attention maps show or what insights we can gain from them. I suspect the "important" neurons in layer 1 are just the ones with the most spikes, which form good basis functions for filling in since their response vectors are (a) dense and (b) tend to have higher signal-to-noise ration due to the (very approximate) Poisson statistics of spike trains.






## Details

 - Fig 3A: Why does performance drop if you increase the ensemble size beyond approx. 8 models? Please also state how many individual models you use to form one ensemble.

 - Paragraph at l. 123: I think it is reasonable to focus on co-bps for the Bayesian hyperparameter search. At the same time I do think that the mask loss – which is probably the Poisson log-likelihood? – correlates with the 4 evaluation metrics you compare it to in Fig 2A. If they did not correlate at all, it would be unclear how training the model with the log-likelihood objective could improve the model’s performance on an uncorrelated evaluation metric.

 - You describe two out of the four datasets you used in Sec 3.1 and 3.2 in detail. I think your paper would improve if you would additionally describe the two other datasets you used. You might be able to cut down on the level of detail you provide in 3.1 and 3.2 in case the pages limit would be a problem.

 - Fig 1: The dimensions of X and X^T are stated wrong. According to line 69 X is N x T, and the spike rasters in Fig. 1 suggest the same. Accordingly, the size of the temporal attention matrix in the figure should be 5 x 5 and the spatial one 3 x 3. For X and X^T the columns should be colored, not the rows.

 - The model is well described, although you could make explicit in the Methods section and maybe Fig. 1 that the model is made of 4 layers of the transformer block depicted in Fig. 1

 - L 107: you could make clear that \tilde{z} is the “output *log* firing rate”

 - L 131: Introduce later used abbreviation NLB

 - L 125 and l 150: you could indicate here that these two metrics correspond to the log-likelihood you use for model training to improve clarity

 - Fig 3C: This Fig might improve if you plot the ground truth and the predicted traces into one plot for better comparability. Please add a description what the different colors mean (different examples?).

 - L 226: instead of “across 4 attention layers” is not quite correct, as you only depict layers 1 and 4. As a matter of fact, it would be interesting to see the same plots for layer 2 and 3.

 - L 258: I do not understand the meaning of the part of the sentence starting “as well as the discovery…”.

 - The paragraph starting at l. 264 is hard to comprehend as the sentences are a bit long and complicated to read. This paragraphs readability could improve a lot by rewording / restructuring it.



**Limitations:**

Yes

**Strengths And Weaknesses:**

## Strengths
 + Research question is important and the presented modeling idea makes sense
 + The paper is well written and easy to understand
 + The motivation in the Introduction section as well as the comparison to related work is decent
 + Compares the model to a variety of baselines + metrics

## Weaknesses
 - New attention mechanism does not lead to measurable improvement
 - Contrastive objective does not lead to measurable improvement
 - Unclear how attention maps provide "interpretability"
 - Some of the methods not clearly described (augementation, masking, positional encodings?)

---

> ### Author Response · Authors · 2022-08-02
> **Response to Reviewer gfnD**
>
> We thank the reviewer for a thoughtful review and valuable feedback. We answer the reviewer's questions and provide revisions to address reviewer's concerns below. We will include revisions in the revised version of the manuscript.
>
> **Re: Questions:**
>
> **Q1.** We would like to emphasize that STNDT with the **inclusion of spatial attention module outperforms AESMTE**, and the improvement is ***not* random** fluctuations. To support this point, we have retrained our best STNDT model and best AESMTE model with five different random seeds and report the mean as well as the standard error in Tables 1-3 below. For AESMTE, we used the same public code and the same set of hyperparameters of the best performing model they provided to ensure a fair comparison. All the results are obtained on the hidden test set held by NLB. The results indicate that STNDT maintains a gap over AESMTE and is more robust across initializations. We will provide these results in the revised manuscript.
>
> We would also like to emphasize that STNDT contributes the most improvements in terms of the primary metric (co-bps). Co-bps is considered by NLB and many other previous works [1-6] as the primary metric to evaluate a Latent Variable Model (LVM) since it is the most suitable metric to reflect the LVM’s ability to characterize coordination of neurons across space and time, and is generalizable to diverse use cases of task behaviors, brain regions and dataset sizes [6]. Other metrics are considered secondary due to their limited capability of measuring LVM’s performance: vel R2 and tp-corr measures ability to decode behavior which is only a coarse reflection of the observed neural dynamics, PSTH R2 assumes the task behavior is structured and repeated, and fp-bps assumes that neural activities can be characterized as an autonomous dynamical system [6].
>
> Notably, the improvement on co-bps of STNDT over AESMTE on all datasets is the direct reflection of the spatial attention’s success. Since the spatial attention module aims to learn the relationship between all (observed and unobserved) neurons in the population at training time, it will leverage this information to infer activities of unobserved neurons based on activities of the observed neurons at testing time, which is exactly what co-bps measures.
>
>
> |                     | MC_Maze        |               |               |               | MC_RTT        |               |               |
> |---------------------|----------------|---------------|---------------|---------------|---------------|---------------|---------------|
> | Methods             | co-bps         | vel R2        | PSTH R2       | fp-bps        | co-bps        | vel R2        | fp-bps        |
> | AESMTE1  | 0.3476±0.0035 | 0.9057±0.0006 | 0.6320±0.0071 | 0.2365±0.0031 | 0.1729±0.0090 | 0.5847±0.0618 | 0.0974±0.0044 |
> | STNDT single w/o CL | **0.3659**±0.0003  | 0.8937±0.0013 | **0.6562**±0.0029 | **0.2446**±0.0014 | **0.1883**±0.0019 | **0.6021**±0.0051 | 0.0958±0.0039 |
> | STNDT single w CL   | **0.3668**±0.0005  | 0.8932±0.0012 | **0.6534**±0.0046 | **0.2447**±0.0009 | **0.1923**±0.0009 | **0.5996**±0.0060 | 0.0932±0.0030 |
>
> Table 1: Performance of STNDT with and without contrastive loss (CL) as compared to AESMTE1 on MC_Maze and MC_RTT datasets
>
> |                     | Area2_Bump    |               |               |               |
> |---------------------|---------------|---------------|---------------|---------------|
> | Methods             | co-bps        | vel R2        | PSTH R2       | fp-bps        |
> | AESMTE1             | 0.2483±0.0096 | 0.8370±0.0175 | 0.5628±0.0423 | 0.1261±0.0080 |
> | STNDT single w/o CL | **0.2717**±0.0011 | **0.8730**±0.0048 | **0.7145**±0.0029 | **0.1435**±0.0019 |
> | STNDT single w CL   | **0.2738**±0.0009 | **0.8720**±0.0020 | **0.7098**±0.0038 | **0.1477**±0.0025 |
>
> Table 2: Performance of STNDT with and without contrastive loss (CL) as compared to AESMTE1 on Area2_Bump dataset
>
> |                     | DMFC_RSG      |                |               |               |
> |---------------------|---------------|----------------|---------------|---------------|
> | Methods             | co-bps        | tp-corr        | PSTH R2       | fp-bps        |
> | AESMTE1             | 0.1795±0.0008 | -0.7297±0.0104 | 0.5584±0.0207 | 0.1597±0.0041 |
> | STNDT single w/o CL | **0.1820**±0.0011 | -0.5210±0.0435 | **0.6080**±0.0015 | 0.1429±0.0059 |
> | STNDT single w CL   | **0.1840**±0.0008 | -0.5148±0.0408 | **0.6097**±0.0071 | 0.1444±0.0095 |
>
> Table 3: Performance of STNDT with and without contrastive loss (CL) as compared to AESMTE1 on DMFC_RSG dataset

---

> > ### Author Response · Authors · 2022-08-02
> > **Response to Reviewer gfnD (cont.)**
> >
> > **Q2**. As described in Q1 above, we have performed additional experiments to measure the robustness of the improvements brought about by contrastive objective across five different random seeds and report the results in Tables 1-3. The results indicate that STNDT with contrastive objective maintains a gap over non-contrastive STNDT, most notably on co-bps, and the improvements are robust to different initializations.
> >
> > **Q3**. STNDT uses positional encodings to encode neuron identity. We used sinusoidal positional encodings as in AESMTE1/3 and add them to the input spike train to use in both temporal and spatial self-attention blocks.
> >
> > **Q4**. We drop individual entries X_ij since this masking method will unify the meaning of masked input for both temporal and spatial self-attention blocks. Unlike the entire rows/columns masking method, this X_ij masked elements are distributed along time and space dimensions evenly regardless whether X or X^T is being used.
> >
> > **Q5**. The two augmented inputs used for contrastive loss are masked in the same manner. Inputs X_ij are selected for masking with a probability of a hyperparameter MASK_RATIO. If selected for masking, X_ij are zeroed with probability controlled by a hyperparamter MASK_TOKEN_RATIO. If X_ij is selected for masking and are not zeroed out, it will be altered (the spike count at X_ij is randomly altered to be a random integer value less than the value of maximum spike count in one bin as recorded in the dataset) with a probability MASK_RANDOM_RATIO. If X_ij escapes all the above probabilities, it will be left unchanged. All the aforementioned ratios are hyperparameters, and together with dropout hyperparameters, they are subject to Bayesian hyperparameter tuning. We have provided details on tuning ranges in the supplementary material and will also include additional model details as the reviewer suggested in Q3, Q4 and Q5 in the revised version of the paper.
> >
> > **Q6**. We appreciate the reviewer’s concern on the interpretability of the attention weights and would like to provide additional clarifications and analyses. The attention maps indicate that there is a consistent subset of neurons whose activities other neurons in the population rely on to infer their firing rates. The identification of these important neurons by STNDT helps us gain insights on which neurons play a vital role in the success of LVMs to discover the underlying population firing rates. We also would like to provide additional analyses on the subset of these important neurons as per the reviewer's suggestion. We examined the spiking activities of the important neurons and confirmed that the important neurons are not the ones with the highest spike counts or the most/least variability in spiking activity. In fact, attention weight of a neuron do not correlate or only correlate weakly to its firing activity statistics, as we show in Table 4 the Pearson's correlation of a neuron’s attention weight with the mean and variance of its spiking activity. All correlation values have p-value < 1e-4. These results indicate that STNDT’s spatial attention has picked up on meaningful population features that are more significant than firing rates of the neurons.
> >
> > |                             | MC_Maze | MC_RTT | Area2_Bump | DMFC_RSG |
> > |-----------------------------|---------|--------|------------|----------|
> > | 𝜌(spikes mean, attn weight) | 0.0164  | 0.2217 | 0.0327       | 0.0852   |
> > | 𝜌(spikes var, attn weight)  | 0.0124  | 0.2189 | 0.0353     | 0.0937   |
> >
> > Table 4: Pearson's correlation between spatial attention weight of a neuron versus mean and variance of its spiking activity.

---

> > > ### Author Response · Authors · 2022-08-02
> > > **Response to Reviewer gfnD (cont.)**
> > >
> > >
> > > **Re: Details:**
> > >
> > > **D1**. The models obtained from Bayesian optimization are ranked by co-bps from the highest to the lowest. Then M firing rate predictions of M models starting from the highest to the lowest models are averaged. The prediction performance of M ensembles depends on the quality of individual models included in the ensembles. As we include more models, i.e. increase ensemble size beyond ~8, we are taking into account predictions of worse models, which could drag the collective performance of the ensemble down. The ensemble size for each dataset is provided in the supplementary materials together with other model details.
> > >
> > > **D2**. We thank the reviewer for raising the important point regarding the correlation of mask loss to the model’s performance that we could have made clearer in the paper. We note that the mask loss, i.e. the Poisson negative log likelihood loss, shown in Figure 2 is the validation mask loss obtained at the final training epoch where the best model is checkpointed. The mask loss does decrease during the training process and is still a good objective to guide the training to a good level of metrics performance. However, after a certain threshold which happens near the end of training, the measured mask loss is no longer a good indication of the model performance as measured by the four metrics. Therefore we opted to perform Bayesian hyperparameters search on the primary metric co-bps.
> > >
> > > **Re: Fig. 3C**: the different colors depict trials of different behavior conditions.
> > >
> > > **Re: L226**: The attention weights in layer 2 and layer 3 look similar to layer 4 and the dispersion of attention weights increases from layer 1 to layer 4. We will include the additional attention visualization of layer 2 and layer 3 in future revision of our paper.
> > >
> > > **Re: L258**: By “discovery of stereotyped features across trials of the same behavior conditions”, we mean the extraction of PSTH pattern in repeated behavior. We thank the reviewer for bringing up the obscurity in writing and will revise the text in our revision of the paper.
> > >
> > > **D3-12**: We thank the reviewer for the helpful recommendations regarding text and figures revisions. We will include revisions as the reviewer suggested in the revised version of the paper.
> > >
> > > [1] Jakob H Macke, Lars Buesing, John P Cunningham, Byron M Yu, Krishna V Shenoy, and Maneesh Sahani. Empirical models of spiking in neural populations. In J Shawe-Taylor, R S Zemel, P L Bartlett, F Pereira, and K Q Weinberger, editors, Advances in Neural Information Processing Systems 24, pages 1350–1358. Curran Associates, Inc., 2011.
> > >
> > > [2] Byron M Yu, John P Cunningham, Gopal Santhanam, Stephen I Ryu, Krishna V Shenoy, and Maneesh Sahani. Gaussian-process factor analysis for low-dimensional single-trial analysis of neural population activity. Journal of neurophysiology, 102(1):614–635, July 2009.
> > >
> > > [3] Chethan Pandarinath, Daniel J O’Shea, Jasmine Collins, Rafal Jozefowicz, Sergey D Stavisky, Jonathan C Kao, Eric M Trautmann, Matthew T Kaufman, Stephen I Ryu, Leigh R Hochberg, Jaimie M Henderson, Krishna V Shenoy, L F Abbott, and David Sussillo. Inferring single-trial neural population dynamics using sequential auto-encoders. Nature methods, 15(10):805–815, October 2018.
> > >
> > > [4] Yuan Zhao and Il Memming Park. Variational latent Gaussian process for recovering single-trial dynamics from population spike trains. Neural Computation, 29(5), May 2017.
> > >
> > > [5] Anqi Wu, Nicholas A Roy, Stephen Keeley and Jonathan W Pillow.Gaussian process based non- linear latent structure discovery in multivariate spike train data. Advances in neural information processing systems, 30:3496–3505, December 2017.
> > >
> > > [6] Felix Pei, Joel Ye, David Zoltowski, Anqi Wu, Raeed H Chowdhury, Hansem Sohn, Joseph E O’Doherty, Krishna V Shenoy, Matthew T Kaufman, Mark Churchland, et al. Neural latents benchmark’21: Evaluating latent variable models of neural population activity. arXiv preprint arXiv:2109.04463, 2021.

---

> > > ### Comment · Reviewer_gfnD · 2022-08-08
> > > **Thanks**
> > >
> > > Thanks for the clarifications. They resolve my questions. My enthusiasm is still somewhat dampened by the facts that
> > >
> > > - the improvement by the contrastive objective is still tiny
> > >
> > > - it's still not clear to me what exactly we learn from the attention weights

---

> > > > ### Author Response · Authors · 2022-08-09
> > > > **Response to Reviewer gfnD's comments**
> > > >
> > > > We thank the reviewer for the comments. We also would like to provide additional clarifications below:
> > > >
> > > > **Re: improvement by the constrative objective:** We would like to clarify that although our main contribution is on the spatio-temporal transformer architecture and the most improvement of our model over the baseline is coming from the incorporation of spatial attention module, the contrastive objective also brings about further improvements compared to our non-contrastive variant. As shown in Tables 1-3 above, STNDT with contrastive loss (CL) further improves STNDT without CL by ~2 SEM on average in terms of the primary metric co-bps across all four datasets.
> > > >
> > > > **Re: what we learn from the attention weights:** From the attention weights we learn that the inferred population response could be explained based mainly on the activity of a small subset of important neurons. One possible application for these identified important neurons could be to monitor and predict the performance of LVMs across multiple recording sessions even before running the LVMs on the data recorded in that session. For example, if on a particular session after spike sorting we find that a portion of these important neurons are lost due to recording instability, we can predict in advance how the LVM will perform based on what portion of the important neurons were lost and apply stabilization/recalibration countermeasures if the predicted performance degrades under acceptable thresholds.

---

> > ### Comment · Reviewer_gfnD · 2022-08-08
> > **What do the error bars depict?**
> >
> > Thank you for running these additional experiments. However, I am somewhat confused by the results. Can you clarify what the error bars depict? If the error bars are indeed as tight as your numbers suggest, why do they deviate so much from the numbers reported in the initial submission? For instance, AESMTE1 is reported on MC_RTT to have a co-bps of 0.1927, whereas your table above reports it as 0.1729 +/- 0.009. Also your own model sometimes seems to be 10 SD (SE?) away from the numbers reported above (e.g. Area2_Bump).

---

> > > ### Author Response · Authors · 2022-08-09
> > > **Re: What do the error bars depict?**
> > >
> > > We thank the reviewer for the good question. The error bar depicts the standard error of the mean (SEM). Based on the performance obtained from the 5 random seeds, SEM is calculated as follows:
> > >
> > > $SEM = \frac{\sigma}{\sqrt{N}} = \sqrt{\frac{\sum_{i=1}^{N}{(x_i - \bar{x})^2}}{(N-1)N}}$
> > >
> > > Where $x_i$ is the performance on seed $i$, $\bar{x}$ is the mean performance, and $N=5$.
> > >
> > > We would like to note that the performance of different models could vary across different Pytorch releases or different platforms, even when using identical seeds. Pytorch also acknowledges this discrepancy in their documentation [1]. In the previous submission, we reported the performance of AESMTE1 (e.g. the co-bps of 0.1927 on MC_RTT) according to the best performance they reported in their work [2]. The AESMTE1 models were trained on their device within a different environment from ours. The STNDT models (e.g. on Area2_Bump) in our previous submission were also trained on our different devices and environment, thus the aforementioned discrepancy could happen.
> > >
> > > Therefore, in the revised tables 1-3 above, to illustrate improvement and robustness of STNDT over AESMTE we already took precautions to ensure the fairest comparison possible among the models by 1) retraining AESMTE1 with their code and their reported best hyperparameters, and 2) retraining our best STNDT models with our previously found best hyperparameters. Both 1) and 2) were done on the same device and using the same Python environment. We believe this will allow the fairest comparison possible among the models, even though it means we had to sacrifice our better performance reported previously (e.g. on Area2_Bump). The new results in tables 1-3 above show that STNDT maintains a positive gap over AESMTE1 in performance, especially in terms of the primary metric co-bps.
> > >
> > > [1] https://pytorch.org/docs/stable/notes/randomness.html
> > >
> > > [2] Darin Sleiter, Joshua Schoenfield, and Mike Vaiana. ae-nlb-2021. https://github.com/agencyenterprise/ae-nlb-2021, 2021.

---

### Official Review · Reviewer_457j · 2022-07-11

**Rating:** 6
**Confidence:** 3
**Soundness:** 3 good
**Presentation:** 3 good
**Contribution:** 3 good

**Summary:**

This paper presents a transformer architecture for modelling neural spike trains. The model considers both the spatial (dependency between the activity of different neurons) and temporal aspects of neural activities. The previous model based on transformers only considers the temporal aspect. The method is evaluated on four benchmark datasets and compared with several baselines, indicating the model's effectiveness.

**Questions:**

Among the four assessed metrics, the model is the best one, mainly from the aspect of co-bps; is this a consequence of hyper-parameters being tuned based on co-bps? Are the hyper-parameters of the baseline models also tuned based on co-bps?


**Strengths And Weaknesses:**

Strengths:
The proposed architecture is intuitive and interesting. It also addressed an important shortcoming with the previous transformer model.

The focus on interpretability is interesting and can potentially lead to important discoveries in neuroscience about the role of each brain region.

The paper is well presented and easy to follow.

Weaknesses:
Some of the architectural details are missing from the paper. For example, how many heads were used? What was the exact architecture for each layer? How is the test performance calculated? Is it based on masked inputs on test data?

---

> ### Author Response · Authors · 2022-08-02
> **Response to Reviewer 457j**
>
> We thank the reviewer for a thoughtful review, valuable feedback and acknowledgement of our work. We provide point by point clarifications and answer questions below.
>
> **Re: number of heads**: As in [1], we used 2 heads for a fair comparison as well as computation efficiency since using multiple heads significantly increases the size of the model and training time.
>
> **Re: architecture of each layer**: Apart from the novel incorporation of spatial attention module, each layer is a standard transformer encoder layer with self-attention block + feedforward block and pre-norm residual connection. We will make sure to include these details in the revised version of our paper.
>
> **Re: how test performance was calculated**: The test metrics are calculated in the same manner as validation metrics but on the hidden test set held by NLB. Formulas for the metrics are detailed in [6] and we have also referenced in our paper. At validation and test time, the inputs to the model are not masked. These procedures follow the same method of NDT for a fair comparison.
>
> We appreciate the reviewer’s comments for inclusion of additional architecture and training details in the main paper. We will make sure to describe the model in more detail in the revised version of our manuscript.
>
> **Re: questions**:
>
> The hyperparameters of the NDT baselines (AESMTE1/3) which we directly compare against are also tuned based on co-bps. We also showed that the co-bps metric correlates well with other metrics (vel R2, PSTH R2, fp-bps) which is why we opted to tune hyperparameters based on co-bps. We will also include a short summary of each metric that was fully described in [6] and an explanation of why co-bps is considered the leading metric for Latent Variable Models (LVM). In particular, we will explain that co-bps is considered by the Neural Latents Benchmark and many other previous works [1-5] as the primary metric to evaluate LVMs since it is the most suitable metric to reflect the LVM’s ability to characterize coordination of neurons across space and time, and is generalizable to diverse use cases of task behaviors, brain regions and dataset sizes [6].
>
> [1] Darin Sleiter, Joshua Schoenfield, and Mike Vaiana. ae-nlb-2021. https://github.com/ agencyenterprise/ae-nlb-2021.git, 2021.
>
> [2] Jakob H Macke, Lars Buesing, John P Cunningham, Byron M Yu, Krishna V Shenoy, and Maneesh Sahani. Empirical models of spiking in neural populations. In J Shawe-Taylor, R S Zemel, P L Bartlett, F Pereira, and K Q Weinberger, editors, Advances in Neural Information Processing Systems 24, pages 1350–1358. Curran Associates, Inc., 2011.
>
> [3] Byron M Yu, John P Cunningham, Gopal Santhanam, Stephen I Ryu, Krishna V Shenoy, and Maneesh Sahani. Gaussian-process factor analysis for low-dimensional single-trial analysis of neural population activity. Journal of neurophysiology, 102(1):614–635, July 2009.
>
> [4] Chethan Pandarinath, Daniel J O’Shea, Jasmine Collins, Rafal Jozefowicz, Sergey D Stavisky, Jonathan C Kao, Eric M Trautmann, Matthew T Kaufman, Stephen I Ryu, Leigh R Hochberg, Jaimie M Henderson, Krishna V Shenoy, L F Abbott, and David Sussillo. Inferring single-trial neural population dynamics using sequential auto-encoders. Nature methods, 15(10):805–815, October 2018.
>
> [5] Yuan Zhao and Il Memming Park. Variational latent Gaussian process for recovering single-trial dynamics from population spike trains. Neural Computation, 29(5), May 2017.
>
> Anqi Wu, Nicholas A Roy, Stephen Keeley and Jonathan W Pillow.Gaussian process based non- linear latent structure discovery in multivariate spike train data. Advances in neural information processing systems, 30:3496–3505, December 2017.
>
> [6] Felix Pei, Joel Ye, David Zoltowski, Anqi Wu, Raeed H Chowdhury, Hansem Sohn, Joseph E O’Doherty, Krishna V Shenoy, Matthew T Kaufman, Mark Churchland, et al. Neural latents benchmark’21: Evaluating latent variable models of neural population activity. arXiv preprint arXiv:2109.04463, 2021.

---

> > ### Comment · Reviewer_457j · 2022-08-07
> > **reply**
> >
> > Thank you for your response.
> >
> > Based on the response and also comments from other reviewers, I'll keep my score.

---

> > > ### Author Response · Authors · 2022-08-09
> > > **Thank you**
> > >
> > > We appreciate the reviewer's evaluation and recognition of our work.

---

### Official Review · Reviewer_uE5F · 2022-07-11

**Rating:** 6
**Confidence:** 4
**Soundness:** 2 fair
**Presentation:** 3 good
**Contribution:** 2 fair

**Summary:**

The paper introduces a spatio-temporal neural transformers: a transformer-based generative model of neural population activity that captures correlations between neurons, in addition to stimulus-driven and temporal correlations in the activity.


**Questions:**

1) It would be good to have a discussion of methods that have been developed to capture noise correlations in neural population acitivity. It would also be good to have a comparison of STNDT to at least one of the methods as well: e.g. Lyamzin et. al. (2010).

2) It would be good to have a more thorough (or at least more measured) discussion of the interpretability of attention weights. Statements as in line 222: "The interpretability...final outcome.";  in line 241 "This finding suggests...not function optimally" and in line 262 "Finally, the novel...not function optimally" may not be warranted when similar analyses on other network architectures exist, with arguably similar results.
It would also be good to have some further analysis on the subset of "important" neurons identified by the attention weights: for example, are these neurons with the highest firing rates, or the least variability in firing rates?

3)  It would be good to have information about training costs, computational and sample efficiency of the menthod

4) Performance of the different methods are compared based on several metrics: it would be good to have some intuition on why these particular metrics were chosen, and also how these are computed.

Minor comments:
1) Figures 1-4 have grey lines around the border that don't nescessarily overlap with the division of subpanels
2) The colourbars in Figure 4 have no label, and the colours are hard to distinguish against the dark background in the plots

**Limitations:**

Overall, although the paper presents an interesting use case of transformers, the value they add over existing methods is not clear. In particular, it is not clear whether the transformers truly are beneficial without a comparison to other methods explicitly set up to capture noise correlations in neural population data. Furthermore, this paper argues for the benefits of STNDT based on the interpretability of the attention weights: however the analysis does not convincingly show whether (a) the attention weights truly pick up on data features more significant than say, the firing rates of the neurons (b) whether they are any more interpretable than weights obtained from linear regression or attribution methods. Finally, without information about the computational costs of this method, it is hard to judge its value, even if it outperforms other methods on the Neural Latents Benchmark tasks.

Taken together, there needs to be more analysis to show whether STNDT can provide more scientific insight and add more value to generative modeling of neural population data, over existing methods.


**Strengths And Weaknesses:**

Overall the paper is well-written, the methods and results are presented clearly. In particular, this seems to be an interesting use-case for transformers, and the ablation study with the contrastive loss is also interesting.

However, there are some major concerns regarding the paper:
1) There is a substantial body of work concerning capturing correlations between neurons (many of which also concurrently capture stimulus-driven /  temporal variability in the population activity -- it would be good to have a discussion of these papers, and also, comparisons of STNDT against some of them. A non-exhaustive list includes:
- Schneidman et al (2006): https://europepmc.org/backend/ptpmcrender.fcgi?accid=PMC1785327&blobtype=pdf
- Macke et al. (2011): http://www.gatsby.ucl.ac.uk/~maneesh/papers/macke-etal-2011-nips-preprint.pdf
- Lyamzin et al. (2010): http://journal.frontiersin.org/article/10.3389/fncom.2010.00144/abstract
- Molano-Mazon et al. (2018): http://arxiv.org/abs/1803.00338
- Ramesh et al (2019): https://openreview.net/forum?id=S1xxRoLKLH
- Bashiri et al. (2021): https://openreview.net/pdf?id=1yeYYtLqq7K

2) While STNDT certainly achieves better performance compared to other methods, it is not clear what scientific insight can be gained from using a transformer. While section 3.3 and 3.4 describe how attention weights can be used to find sub-networks of "important" neurons and model consistency, it is not clear whether these weights are any more interpretable than say, regression weights, or those computed on CNNs with attribution methods. Indeed, attribution methods such as GradCAM (https://arxiv.org/abs/1610.02391) and axiomatic attribution (https://arxiv.org/abs/1703.01365) have been used in conjunction with other deep neural network architectures trained on neural population activity in precisely the same manner described here, with arguably similar success (e.g. Maheswaranathan et al. 2018: https://www.biorxiv.org/content/early/2018/06/08/340943).

3) It would also be nice to have an estimate of the training costs, computational and sample efficiency of STNDT in comparison with other methods -- otherwise it is hard to truly judge the benefits of using this method, over others for datasets beyond those described in the paper.

---

> ### Author Response · Authors · 2022-08-02
> **Response to Reviewer uE5F**
>
> We thank the reviewer for a thoughtful review, valuable feedback and additional references. We provide point by point clarifications and answer questions below.
>
> **W1**. We appreciate the reviewer bringing these works to our attention and will make sure to cite and discuss all of these references in our revised manuscript. While these works were developed to capture noise correlations in the neural population activities and could resemble our work in the motivations, there are key differences that distinguish our approach from these methods, especially on the goals that influenced our design choice. We elaborate on these below and will include this discussion in the future revised version of the manuscript.
>
> - **Explicit vs. Implicit conditioning on stimulus/behavior**: Many of the previous works attempting to model interactions of neurons in the population [1-5] fitted the model separately for each unique stimulus/behavior, thus learning the neural interaction that is restricted to the examined stimulus/behavior only. In contrast, STNDT is trained with multiple behavior conditions and aims to learn rich covariation of neurons encompassing all recorded behavior conditions.
>
> - **Reliance on external variables vs. Self supervision**: By fitting the model with respect to each stimulus [1-5], or with thousands of stimuli [6], previous methods rely on the knowledge of which stimulus/condition the neural responses belong to. STNDT, on the other hand, is trained in an unsupervised manner and learns to model the interaction among neurons without access to any other observation apart from the population spiking activity.
>
> - **Generation of spike trains vs. Capturing the (denoising) firing rates**: While the goal of modeling neuron interaction in existing methods is to aid with the generation of realistic neural activities associated with the induced stimuli [1-5], in STNDT it is to better infer the firing rates underlying the noisy spiking activity.
>
> - **A priori assumptions vs. Assumption-free modeling**: Unlike [3] and similar to [4], STNDT’s modeling of relationship among neurons does not rely on any explicit assumption regarding the statistics of neural activity, e.g. assumption that noise correlation is constant across time bins and trials and needs to be matched by the model.
>
>
> **W2**. We thank the reviewer for bringing up the resemblance of our spatial attention weights interpretability with existing works on interpretability of deep learning models. We acknowledge the relevancy and will cite these works along with a discussion in the revision of our manuscript. In particular we will discuss the following differences:
>
> - [7] and [8] highlight the contribution of different components in the input to the overall output class prediction, such as pixels in an image of a dog to the ‘dog’ probability prediction. However, it is unclear whether these methods could be applicable in an **unsupervised setting** as in our work.
>
> - The notion of interpretability in [9] is built upon the receptive fields of CNN units which are then compared to that of biological retinal interneurons. The method thus revolves around the CNN structure, which is relevant in relating visual stimuli to neural response in their investigation, but is unclear if the CNN structure is suitable to **non-visual unsupervised settings** such as the spatiotemporal denoising of neural activities in our work.
>
> - As noted in [8], the relative importance of input features attributed to the model prediction is computed independently and does not take into account the interactions between input features. In contrast, the interpretability of our attention weights is built upon **how the activities of other neurons influence one neuron** in obtaining its prediction, thus interactions among model inputs are especially leveraged and taken into account.
>
>
> **W3**. We have provided information regarding training and computational efficiency of our model in the Supplementary Material. We thank the reviewer for emphasizing the importance of this information and will make sure to include more training details in the main paper and refer readers to the supplementary material for additional details. We will also provide additional information on the sample efficiency of our model, which is directly connected to the full sizes of the four datasets in the Neural Latents Benchmark (NLB) detailed in [10].

---

> > ### Author Response · Authors · 2022-08-02
> > **Response to Reviewer uE5F (cont.)**
> >
> > **Re: Questions**:
> >
> > **Q1**. Please see the discussion in W1 that we will also include in the future revised version of the manuscript.
> >
> > **Q2**. We appreciate the reviewer’s suggestion of a more thorough discussion on the interpretability of spatial attention weights in comparison to other works on interpretability of deep learning models and have addressed it in the W2 comment above. We also would like to provide additional analyses on the subset of important neurons identified by our STNDT’s spatial attention weights. We have examined the spiking activities of the important neurons and confirmed that the important neurons are not the ones with the highest spike counts or the least variability in spiking activity. In fact, attention weight of a neuron do not correlate or only correlate weakly to its firing activity statistics, as we show in Table 1 the Pearson's correlation of a neuron’s attention weight with the mean and variance of its spiking activity. All correlation values have p-value < 1e-4. These results indicate that STNDT’s spatial attention has picked up on meaningful population features that are more significant than firing rates of the neurons.
> >
> > |                             | MC_Maze | MC_RTT | Area2_Bump | DMFC_RSG |
> > |-----------------------------|---------|--------|------------|----------|
> > | 𝜌(spikes mean, attn weight) | 0.0164 | 0.2217 | 0.0327 | 0.0852 |
> > | 𝜌(spikes var, attn weight)  | 0.0124 | 0.2189 | 0.0353| 0.0937 |
> >
> > Table 1: Pearson's correlation between spatial attention weight of a neuron versus mean and variance of its spiking activity.
> >
> > **Q3**. Please see the discussion in W3 that we will also include in the future revised version of the manuscript.
> >
> > **Q4**. Four evaluation metrics are proposed and evaluated by NLB to have a fair comparison among the competing approaches. We have pointed readers to the NLB paper for more details of the reasons for choosing these metrics as well as how the metrics are computed. We acknowledge that the clarity of our method could be improved if this information is also discussed in the main paper and we will provide a short summary in our revision of the paper.
> >
> > Co-bps measures the ability of LVMs to infer activities of held-out neurons from the training neurons at test time. Co-bps is chosen based on the observation that latent representations are distributed across many neurons in the population [10]. Co-bps reflects the LVM’s ability to characterize coordination of neurons across space and time, and is generalizable to diverse use cases of task behaviors, brain regions and dataset sizes [10]. It is computed as follows:
> >
> > $\text{co-bps}=\frac{1}{n_\{\text{sp}}\text{log}2}(L(\lambda\_{n,t};\hat{y}\_{n,t})-L(\bar{\lambda}\_{n,:};\hat{y}\_{n,t}))$
> >
> > Where $\lambda\_{n,t}$ is the predicted firing rate of held-out neuron $n$ at time point $t$, $\bar{\lambda}\_{n,:}$ is its mean firing rate, $\hat{y}\_{n,t}$ is its spike count at time point $t$, $L$ is the sum of Poisson log-likelihood over all held-out neurons and time points, $n\_{sp}$ is the total number of spikes. A positive co-bps indicates that the model infers the time-varying activity of held-out neurons better than the flat mean firing rate.
> >
> > Vel $R^2$ in MC_Maze, MC_RTT, Area2_Bump datasets measures how much the LVM helps in decoding the behavior. It is chosen based on the assumption that a good LVM should be able to relate latent states with behavior. First a Ridge regression is done to fit the training rates to the correponding hand velocity, then $R^2$ is computed between the velocity prediction and the ground truth velocity.
> >
> > Tp-corr is the metric to measure behavior decoding in the DMFC_RSG dataset. It was observed that the rate of change in the population state (neural speed) correlates negatively with the time $t_p$ between the Set cue and the monkey’s Go response in the Ready-Set-Go task. Tp-corr is computed by first calculate the average neural speed from the test rate prediction and then compute the Pearson’s correlation between the neural speed and the measured $t_p$.
> >
> > PSTH $R^2$ measures the match of LVM’s prediction to Peri-Stimulus Time Histogram (PSTH) in tasks with repeated trials. It is chosen based on evidence that LVMs can capture stereotyped features of neurons’ responses in repeated trials. For each neuron, we calculate $R^2$ between predicted rates on all trials and the PSTHs for the conditions corresponding to those trials, then take the average across all neurons to get the final PSTH $R^2$. Each PSTH is computed while excluding the trial being evalated.
> >
> > Fp-bps measures LVM’s ability to predict responses of all neurons at unseen future time points. It is chosen based on the assumption that future neural activity can be predicted based on prior neural activity, and is best applied in cases where neural activity can be characterized as an autonomous dynamical system. Fp-bps is computed in the similar manner as co-bps but on the held-out time points of all neurons.

---

> > > ### Author Response · Authors · 2022-08-04
> > > **Response to Reviewer uE5F (cont.)**
> > >
> > > **Re: Minor comments:**
> > >
> > > **C1**. We did not observe the mentioned grey lines around the borders on our end. This could arise from differences between PDF readers. We thank the reviewer for reporting this phenomenon and will make sure to test our revised paper rendering on different platforms.
> > >
> > > **C2**. We appreciate the reviewer’s suggestions and will add labels for the colorbars as well as adjusting the colormap for better visual experiences.
> > >
> > > [1] Schneidman, Elad, et al. "Weak pairwise correlations imply strongly correlated network states in a neural population." Nature 440.7087 (2006): 1007-1012.
> > >
> > > [2] Macke, Jakob H., et al. "Empirical models of spiking in neural populations." Advances in neural information processing systems 24 (2011).
> > >
> > > [3] Lyamzin, Dmitry R., Jakob H. Macke, and Nicholas A. Lesica. "Modeling population spike trains with specified time-varying spike rates, trial-to-trial variability, and pairwise signal and noise correlations." Frontiers in computational neuroscience 4 (2010): 144.
> > >
> > > [4] Molano-Mazon, Manuel, et al. "Synthesizing realistic neural population activity patterns using generative adversarial networks." arXiv preprint arXiv:1803.00338 (2018).
> > >
> > > [5] Ramesh, Poornima, Mohamad Atayi, and Jakob H. Macke. "Adversarial training of neural encoding models on population spike trains." (2019).
> > >
> > > [6] Bashiri, Mohammad, et al. "A flow-based latent state generative model of neural population responses to natural images." Advances in Neural Information Processing Systems34 (2021): 15801-15815.
> > >
> > > [7] Selvaraju, Ramprasaath R., et al. "Grad-cam: Visual explanations from deep networks via gradient-based localization." Proceedings of the IEEE international conference on computer vision. 2017.
> > >
> > > [8] Sundararajan, Mukund, Ankur Taly, and Qiqi Yan. "Axiomatic attribution for deep networks." International conference on machine learning. PMLR, 2017.
> > >
> > > [9] Maheswaranathan, Niru, et al. "Deep learning models reveal internal structure and diverse computations in the retina under natural scenes. bioRxiv." URL: https://www. biorxiv. org/content/early/2018/06/14/340943. http://dx. doi. org/10.1101/340943. arXiv: https://www. biorxiv. org/content/early/2018/06/14/340943. full. pdf (2018).
> > >
> > > [10] Felix Pei, Joel Ye, David Zoltowski, Anqi Wu, Raeed H Chowdhury, Hansem Sohn, Joseph E O’Doherty, Krishna V Shenoy, Matthew T Kaufman, Mark Churchland, et al. Neural latents benchmark’21: Evaluating latent variable models of neural population activity. arXiv preprint arXiv:2109.04463, 2021.

---

> > > > ### Comment · Reviewer_uE5F · 2022-08-07
> > > > **Thank you**
> > > >
> > > > I thank the authors for their detailed clarifications, particularly with respect to related work on modeling noise correlations in neural population data, and also for the extra analysis checking whether the attention weights correspond to the mean firing rate of the neurons.
> > > >
> > > > It would be interesting to see whether the attention weights are correlated with any other feature of the spiking activity -- I am still concerned about interpretability of the weights, in the sense that we cannot be certain which features of the population activity they are picking up on. However, in the absence of further analysis, it would at least be good to have a sentence or two addressing this point in the discussion (or if Table 1 from the responses in included in the manuscript, then in the same section as the table).
> > > >
> > > > Nevertheless, I thank the authors, once again for their detailed explanations, and have increased my score to 6.

---

> > > > > ### Author Response · Authors · 2022-08-09
> > > > > **Response to Reviewer uE5F's comments**
> > > > >
> > > > > We thank the reviewer for the recognition of our work. We will include the discussion as the reviewer suggested. Although the question of what non-trivial input features (other than trivial mean spiking activity) the spatial attention picked up on is an interesting future work, one could immediately think of ways how such important neurons could help us practically. One possible application for these identified important neurons could be to monitor and predict the performance of LVMs across multiple recording sessions even before running the LVMs on the data recorded in that session. For example, if on a particular session after spike sorting we find that a portion of these important neurons are lost due to recording instability, we can predict in advance how the LVM will perform based on what portion of the important neurons were lost and apply stabilization/recalibration countermeasures if the predicted performance degrades under acceptable thresholds.

---

### Meta-Review · Area_Chair_DEyu · 2022-08-30

**Recommendation:** Accept
**Confidence:** Less certain

**Metareview:**

This paper introduces a spatiotemporal neural data transformer (STNDT) model for multineuronal spike trains. The method achieves state of the art performance on a variety of tasks within the Neural Latents Benchmark (NLB), which was introduced at NeurIPS in 2021. Technically speaking, the approach is a slight tweak on standard transformers, but it allows for trial-by-trial reweighting of values based on a neurons-by-neurons similarity matrix. It is interesting that this tweak is enough to boost performance.

While the reviewers expressed some hesitation with regard to the degree of technical novelty and the questionable interpretability of the model, I think it is important to recognize improvements in benchmarks that the field has put forward at NeurIPS. The paper is well written and the results are thorough. Overall, I think it is a valuable contribution.

In addition to addressing the reviewers' concerns, I would like the authors to address the following points in the final paper:
- I still think Figure 1 is confusing. The spike train in the top row and the firing rates in the bottom left seem to suggest that there are N=3 neurons and T=5 time steps, but the matrices and stated shapes suggest that there N=5 neurons and T=3 time steps. I believe the authors intended to have N=3 and T=5. If so, please correct the figure to help avoid any confusion!
- The introduction does a good job of summarizing related work in statistical neuroscience, but it does not, as far as I can see, discuss the enormous literature on variations of transformers. I do not know the best references, but I would be surprised if there haven't been related works in the transformer literature for other spatiotemporal tasks (e.g. applied to modeling audio or video data). The lack of discussion of related work in the broader ML community is a glaring omission.

**Award:**

No

---

### Decision · Program_Chairs · 2022-09-14

Accept